# Analyses of Pepper Cinnamoyl-CoA Reductase Gene Family and Cloning of CcCCR1/2 and Their Function Identification in the Formation of Pungency

**Dan Wu** [1,†], **Miao Ni** [1,†], **Xin Lei** [1,†], **Liping Zhang** [1], **Wei Zhang** [1], **Huangying Shu** [1], **Zhiwei Wang** [1], **Jie Zhu** [1], **Shanhan Cheng** [1,*], **Pingwu Liu** [1], **Honghao Lv** [2] and **Limei Yang** [2]

1    College of Horticulture, Hainan University, Haikou 570228, China; cocoapuff_93@163.com (D.W.); rockymiaoni@163.com (M.N.); l1031667827@163.com (X.L.); zlp602900665@163.com (L.Z.); zhang914wei@163.com (W.Z.); hnhyshu@163.com (H.S.); zwwang22@163.com (Z.W.); jasperjie@aliyun.com (J.Z.); hnulpw@dingtalk.com (P.L.)

2    Institute of Vegetables and Flowers, Chinese Academy of Agricultural Sciences, Beijing 100098, China; lvhonghao@caas.cn (H.L.); yanglimei@caas.cn (L.Y.)

\*    Correspondence: 990865@hainanu.edu.cn; Tel.: +86-131-3893-9608

†    These authors contribute equally to this work.

**Abstract:** Cinnamoyl-CoA reductases (CCR) have a possible role in pungency formation of pepper because they can convert feruloyl-CoA, sinapoyl-CoA, and p-coumaroyl-CoA into lignin, which are also competitive precursors of capsaicin biosynthesis in phenylpropanoid metabolism. In this study, genome-wide *CCR* gene family, exon–intron structures, sequence homology, phylogenetic characterization, and promoters were analyzed in pepper. Two *CCR* genes were cloned from *Capsicum chinense*, their enzymic kinetic parameters and regulatory function were identified by heterologous expression, ectopic expression, and VIGS. In total, 38 genes were found as predicted CCRs or CCR-like proteins and were composed of 2–10 exons. The promoters of pepper *CCRs* contained growth, stress, hormone, and light-response elements. The affinity and catalytic efficiency of CcCCR1/2 to feruolyl-CoA was the highest. The analysis of metabolic substances showed that capsaicin content was negatively correlated with lignin and positively correlated with flavonoids. The highest expression of *CcCCR1* was found in stems, the higher expression of *CcCCR2* was found in stem and early fruit than other organs. *CCR1, 2* had certain effects on capsaicin content by regulating related enzyme activity, *CCR2* played a more important role in regulating pungency formation. Our results clarify the competitive mechanism between lignin and capsaicin biosynthesis and provide an explanation for spice regulation.

**Keywords:** *Capsicum chinense* Jacquin; CCR family; capsaicin; enzymatic kinetics; pungency regulation

## 1. Introduction

Pepper (*Capsicum* spp.) has more than 3.7 million hectares growing areas all over the world [1], and is consumed mainly as fresh vegetables and food additives because of its unique color, spice, and flavor. The pungent compounds are a group of alkaloids named capsaicinoids (Caps), which include capsaicin, dihydrocapsaicin, and more than 10 other spice ingredients, and are produced uniquely by the Capsicum genus [2]. Caps not only have a pharmacological effect and can be used to relieve aching, eliminate inflammation, regulate appetite, and control rheumatism, but also have the function of preventing fruits from being attacked by animals or pathogens [3]. In addition, Caps are also an important substance of chili water used by police to prevent and as defense against violence [4].

Capsaicin accounts for 69% of the total capsaicinoid content and is exclusively synthesized by the phenylpropanoid metabolic pathway (PMP) and branch chain fatty acid metabolic pathway (BCFMP) in the placental tissue of pepper fruit and is accumulated in

vacuoles of placental epidermis cells [5–8]. In the PMP, the phenylalanine is catalyzed to form 4-coumaroyl-CoA by PAL, C4H, and 4CL successively. Then, 4-coumaroyl-CoA was catalyzed by HCT, C3H and COMT, pAMT to form caffeic acid, ferulic acid, vanillin, to vanilylamine. In the BCFMP, pyruvate is catalyzed by ALS, AHRI, DHAD, BCKDH, BCAT, KAS, ACL, FAT, ACS, and finally converted into 8-methyl-6-solenic acid CoA. Then, capsaicin is synthesized through the condensation of vanilylamine and 8-methyl-6-nonlnoyl-CoA catalyzed by CS (Figure S1).

As the most important secondary pathways in plants, the PMP intermediate products including p-coumaroyl-CoA, caffeoyl-CoA, and feruloyl-CoA are the likely competitive precursors for synthesis of capsaicin, flavonoids, isoflavones, alkaloids, lignin, and coumarin in plants (Figure S2) [9,10]. In previous research, capsaicin synthesis is promoted and the contents of lignin and flavonoids are reduced simultaneously in pepper under the condition of different light intensity, but the level of tannin had no significant change [11]. The highest content of capsaicin and lowest content of lignin are found in the placental tissue at 40 days after flowering (DAF) of pepper treated with suitable N concentration [12]. Moreover, the content level of capsaicin is enhanced and the formation of lignin is contrary in putrescine treatment in pepper [13]. The soluble phenolic and lignin contents in the pepper fruits of control plants without mineral supplementation were found to be higher than those of mineral supplemented plants, and mineral supplementation is favorable to increase the capsaicinoid contents [14]. The negative correlation between lignin and capsaicin content is also found in other research [15]. However, the specific cause of the impact of the lignin biosynthesis pathway on the capsaicin content in the PMP, and the molecular mechanism of the competitive relationship between lignin and capsaicin, are still unclear in the placental tissue of peppers.

Cinnamoyl-CoA reductase (CCR; EC 1.2.1.44) catalyzes the conversion of cinnamoyl-CoAs into their corresponding cinnamaldehydes and represents the enzymatic entry step that leads into the monolignol-specific branch of the PMP in angiosperms [16]. This enzyme may regulate the overall carbon flux towards lignin production and balance the abundance of phenolic compounds. Suppression of CCR in angiosperms changed the metabolite profile and resulted in lignin reductions of up to 50%, providing evidence for the crucial role of CCR in monolignol biosynthesis [17,18]. Homologs of *CCR* and *CCR like* gene families have been reported to be diverse in plant species, including 11 genes in *Arabidopsis thaliana*, 9 in *Populus trichocarpa*, 33 in *Oryza sativa*, and 10 in *Eucalyptus grandis* [19–25]. Functional studies of CCRs have been performed in some plant species including *A. thaliana*, *Eucalytus gunnii*, *Medicago truncatula*, *Petunia hybrida*, *Paspalum dilatatum*, *Populas euramericana*, *Zea mays*, *Panicum virgatum*, and *Triticum aestivum* [19–25]. In plants, CCRs are mainly divided into two classes by their structure and function. CCR1 and CCR2 are phylogenetically distinct, differentially expressed, and the corresponding enzymes exhibited different biochemical properties with regard to substrate preference. CCR1 exhibits preference for feruloyl CoA, while CCR2 prefers caffeoyl and 4-coumaroyl CoAs, and both of them exhibit sigmoidal kinetics with these substrates [26,27]. However, the family of *CCR* genes in pepper and their enzyme kinetic characteristics are still unknown.

In previous works, Mazourek et al. speculated that *CCRs* were probably candidate genes regulating capsaicin biosynthesis [8]. Several CCRs were found to be differentially expressed in the transcriptome data of placental tissue at different developmental stages of pepper fruits, and their expression level was negatively correlated with the content of capsaicin. [28]. It can be seen that *CCR* genes of pepper play a regulatory role in the formation of pepper pungency. In this study, the structure and evolutionary relationship of *CCR* and *CCR-like* gene families were analyzed, and the full-length CCR cDNA were cloned from Huangdenglong pepper (*Capsicum chinense* Jacquin), a particularly hot pepper in Hainan, and their gene sequences and enzymatic characteristics were identified. Furthermore, the function of CCRs regulating lignin content and spicy flavor was confirmed to establish the competitive relationship between the formation of spicy substances in pepper and the

synthesis of related metabolites and analyze the molecular mechanism of the regulation of capsaicin metabolism in the lignin branch of PMP.

## 2. Materials and Methods

### 2.1. Plant Materials and Experimental Reagents

Huangdenglong peppers with high pungency were used for cloning the *CCR* genes, and another two varieties including Haijiao 218 (Hainan Jiangbao Seed Co., Ltd., Haikou, China) with medium pungency, Zhongtian F1 (Zhanjiang Earth Vegetable Seed Co., Ltd., Zhanjiang, China) with no pungency, were used to analyze the content of metabolites. All materials were planted in the horticultural experimental station of Hainan University. p-Coumaric acid, ferulic acid, sinapic acid, coenzyme-A (CoA), and reduced b-nicotinamide adenine dinucleotide phosphate (NADPH) for hydroxycinnamoyl-CoA production were purchased from Sigma-Aldrich. Reagents for buffers, media, and other solutions were obtained from Sigma-Aldrich (St. Louis, MO, USA), Shenggong (Shanghai, China), Vazyme (Nanjing, China), Suzhou Comin Biotech Co., Ltd. (Suzhou, China), etc.

### 2.2. Multiple Sequence Alignments, Phylogenetic Analysis of Pepper CCRs and CCR-like Gene Family

Deduced protein sequences of pepper CCRs and CCR-like gene family, and functional CCRs identified from other plant species were retrieved from the National Center for Biotechnological Information database (https://www.ncbi.nlm.nih.gov/ (accessed on 16 January 2022)). Multiple CCRs sequence alignment was performed using Clustal Omega (https://www.ebi.ac.uk/Tools/msa/clustalo/ (accessed on 16 January 2022)). The alignment result was further reformatted with Mview (https://www.ebi.ac.uk/Tools/msa/mview/ (accessed on 16 January 2022)). In MEGA 6 software, the neighbor-joining (NJ) method was used to undertake phylogenetic analysis.

### 2.3. Analysis of Cis-Regulatory Elements in Pepper CCR and CCR-like Gene Family

We analyzed the 2000 bp sequence upstream from each gene's initiation codon (ATG) in the pepper genome and evaluated the type and number of cis-acting regulatory DNA elements using the PLANT CARE program (http://bioinformatics.psb.ugent.be/webtools/plantcare/html/ (accessed on 16 January 2022)).

### 2.4. Cloning and Bioinformatics Analysis of CcCCR1 and CcCCR2 from Huangdenglong Pepper

The young stem tip of Huangdenglong pepper was preserved at −80 °C. The total RNA was extracted by Trizol reagent (Invitrogen Company, Waltham, MA, USA) and cDNA was synthesized by PrimeScriptTM RT reagent kit (Takara, Dalian, China). The product was stored at −20 °C. The primers (Table S1) are designed to clone the CCR gene according to homologous sequences in NCBI and analyze the expression of related genes. The intermediate fragment of *CCR1* and *CCR2* was cloned, and then the 5′ and 3′ ends of *CCRs* were cloned by SMARTer ®RACE/5′, 3′ kit (Takara, Dalian, China). The PCR products were purified and cloned into the pMD18-T vector (Takara, Kusatsu, Japan), and propagated in *E. coli* DH5α (Tianyihuiyuan Co., Ltd., Beijing, China), and inserts were confirmed by sequencing. Sequences were deposited at NCBI Genbank with accession OL660766 and MN548390.

### 2.5. Expression of CCR1 and CCR2 in E. coli and Their Enzyme Kinetics Parameters

A pair of specific primers containing NdeI and BamHI enzyme sites were designed by primer 5. PCR amplification was performed by using pMD18-*CcCCRs* as the template during gene cloning. *CcCCRs*-pET28(a) was transformed to TOP10 (Tiangen, Beijing, China) to obtain recombinant protein containing the N-terminal His tag. In LB medium at 37 °C, transformed bacteria were cultured to $OD_{600} = 0.6$. Expression was induced with 0.1 mM IPTG, incubated at 30 °C for 8 h, and centrifuged at 8000 rpm for 3 min. The cell NTA-0 buffer was heavy and the bacteria were precipitated by ultrasound. The precipitation was

suspended in 50 mL NTA-0 buffer, and the impurity protein dissolution was promoted by adding DTT to the final concentration of 1 mM using ultrasound. Then, the sample was centrifuged, and the steps repeated until the upper clearance was transparent. The PBS heavy suspension was precipitated, ultrasonic bacteria broken the same as above, centrifugation was performed to 6 m guanidine hydrochloric acid heavy suspension inclusion body, and DTT was added to the final concentration of 5 mM. Concussion dissolution included culvert, centrifugation, and extraction of SDS-PAGE (DYCZ-20G, Beijing Liuyi Co. Ltd., Beijing, China) detection. The protein solution was diluted with 3 M guanidine hydrochloride and stirred in pH8.0 renaturation solution for 24 h. The protein solution was placed in a dialysis bag and concentrated to 50~100 mL by PEG 20,000. TE buffer dialysis was performed overnight. The protein solution was concentrated with PEG 20,000 to $2 \leq 4$ mL. Dialysis was performed at 4 °C with TE buffer overnight.

The albumin solution was filtered by a 0.22 μm filter. An Ni-NTA column was used to sample the albumin solution at a flow rate of 1 mL/min. NTA-0 buffer (pH8.0) was used to wash the column with the flow G250 detection solution without discoloration. The eluents were eluted with 20, 60, 200, and 500 mM imidazole, respectively, and the eluents were collected in G250 detection solution without discoloration. Deionized water was used to wash the column and 20% ethanol sealed the column. Eluent was detected by SDS-PAGE electrophoresis.

In this experiment, the enzymatic characteristics of CcCCR1 and CcCCR2 protein were determined by establishing an enzyme reaction system. The reaction system was as follows: phosphate buffer of 50 mM, pH 6.25, 0.5 mM NADPH, 0.1–0.3 mM substrate, and 500 ng recombinant CcCCRs protein, the total volume of which was added at 0.5 mL, 30 °C, and detected by UV spectrophotometer (T6, Beijing PuXi Co. Ltd., Beijing, China). When the substrate was feruloyl-CoA, sinapoyl-CoA, *p*-coumaroyl-CoA, the optimum absorption peaks were 346, 350, and 333 nm, respectively. The consumption of substrate (control substrate standard curve) and the activity of reactive enzyme were measured [29–32].

The reaction system was placed at 10–80 °C, 10 °C as a gradient for 20 min, and the absorbance value at 346 nm was determined by adding protease, the reaction system was repeated three times [30]. The reaction system was placed in a phosphoric acid buffer at pH 4.58, with a gradient of 0.5 at intervals, and the changes in absorbance values at 346 nm were determined by the addition of protease and were repeated for every three treatments [30]. Feruloyl-CoA, sinapicoyl-CoA, and *p*-coumaroyl-CoA were purchased from Yuanye Shanghai.

### 2.6. Heterologous Expression of CcCCR1 and CcCCR2 Genes in Arabidopsis thaliana

The full-length ORF of *CcCCR1* and *CcCCR2* was cloned, and the pBI121-*CcCCRs* expression vector was constructed and introduced into Agrobacterium GV3103. *Arabidopsis thaliana* was infected by Agrobacterium dipping method three times. The harvested T0 generation seeds were sown on demand in 1/2MS medium (40 μg/mL Kan) containing antibiotics to screen the normal-growth-positive seedlings. T1 generation seeds were harvested, and plants were screened and cultured for PCR detection. T2 generation homozygous Arabidopsis which obtained *CcCCR1* and *CcCCR2* genes was screened for q-PCR detection and lignin content was determined.

### 2.7. Virus-Induced Gene Silencing of CcCCR1 and CcCCR2 in Hainan Huangdenglong Pepper

The pTRV vector and *Agrobacterium tumefaciens* strain GV3103 were prepared for VIGS. According to the full-length sequences of *CcCCR1* and *CcCCR2* genes, two pairs of specific primers (BamHI and SacI digestion sites) were designed to amplify the target fragments of 353 and 367 bp (Supplementary Table S2 for primers). Then, two target fragments were cloned into the pTRV2 vector by OneStep Cloning Kit to yield the pTRV2: *CcCCR1* and pTRV2: *CcCCR2* construct (Fig10a). The constructed pTRV2: *CcCCR1* and pTRV2: *CcCCR2* expression vectors, and pTRV1 and pTRV2 empty plasmids were transformed into *Agrobacterium tumefaciens,* respectively.

A 10 mL culture of each *Agrobacterium tumefaciens* solution containing pTRV1, pTRV2, and pTRV1 and pTRV2: *CcCCRs*, respectively, was incubated for 24–36 h at 28 °C in LB containing 50 mg·L$^{-1}$ kanamycin, 50 mg·L$^{-1}$ Gentamicin, and 50 mg·L$^{-1}$ rifampicin. The primary culture was resuspended into induction medium containing 50 mg·L$^{-1}$ kanamycin, 20 mg·L$^{-1}$ rifampicin, 50 mg·L$^{-1}$ Gentamicin, 200 µM acetosyringone, and was shaken at 28 °C for 20–24 h until the OD600 value was about 0.5–0.8. The cells were precipitated by centrifugation for 10 min at 3500× *g* and resuspended in the same volume of Agrobacterium infiltration buffer containing 10 mM MgCl$_2$ and 10 mM MES at pH 5.7 until the OD600 reached 1.0. *A. tumefaciens* GV3103 containing pTRV1 was mixed with GV3103 containing either pTRV2 or pTRV2: *CcCCRs* in a 1:1 ratio. After mixing them, 400 µM acetosyringone was added.

The young fruits, about 5 days after flowering, were used for VIGS. About 0.2 mL of infection solution was absorbed with a 1ml sterile syringe and the pepper was injected by oblique pricking from the fruit stem in the young fruit stage (Figure S3). The inoculated plants were grown at 18 °C for 48 h in 60% relative humidity in the dark and then placed in a growth room at 25 °C with a 16 h light/8 h dark photo period.

*2.8. Determination of Relative Expression of CCR Genes and Other Genes Involving in Capsaicin Biosynthesis*

The actin gene (GQ339766.1) of pepper was used as the internal parameter, and the designed primers for pungency-related genes are preswnted in Table S2. The results were analyzed by 2$^{-\Delta\Delta ct}$ method and repeated three times.

*2.9. Determination of Capsaicin, Lignin, and Flavonoids in Peppers*

The fruits of peppers blooming at the same time were stored at −80 °C for 10, 20, 30, 40, and 50 days, respectively, and the stems and leaves on the secondary branches were preserved at −80 °C for 10, 20, 40, and 50 days, respectively. The contents of capsaicin, lignin, and flavonoids were determined by HPLC [28]. Three side holes were set up for each sample. The standard curve was made after the experiment, and the concentration of each sample was calculated.

**3. Results**

*3.1. The CCR Gene Family and Their Intron–Exon Structures in Capsicum L.*

*CCRs* belong to the mammalian 3b-hydroxysteroid dehydrogenase (HSD)/plant dihydroflavonol reductase (DFR) superfamily [16,26]. In the pepper genome database of NCBI, 38 genes were annotated as predicted CCRs or CCR-like (CCR/DFR/epimerase 3b-HSD) proteins, including 21 genes in *Capsicum annuum* (CA), 8 genes in *Capsicum baccatum* (CB), and 9 genes in *Capsicum chinense* (CC), respectively (Table 1). Based on the similarity of the functional CCR open reading frame and peptide length from Arabidopsis, corn, wheat, switchgrass, and *E. gunnii*, respectively [16,19,21,33,34], 18 *CaCCRs* had ORFs of a comparable size (942–1149 nucleotides) to known functional *CCR* genes, which encode 313–382 amino acids (Table 1), indicating that there is no C-terminal extension in *CaCCRs*. Of the 8 CbCCRs, 6 CbCCRs belong the above range of ORF length; however, the length of the other two CCR peptides were 394aa and 451aa, respectively, indicating that there is a C-terminal extension in CbCCRs. Of the 9 CcCCRs, 2 CcCCRs had short ORFs encoding <229 amino acids, and lacked one or both conserved regions. The majority of pepper CCRs are predicted to have an acidic pI and a MW of 34–37 kDa (Table 1).

**Table 1.** Pepper *CCR* and *CCR-like* gene family. [a] [a] pepper genes annotated as CCRs and CCR-like (CCR/DFR/epimerase 3β-HSD) proteins were retrieved from the MSU RGAP database. [b] ORF, open reading frame; [c] MW, molecular weight.

| ID | Name | Gene Description | ORF [b] | Protein Size [c] | Theoretical MW [c] (kDa) | pI | Chromosome |
|---|---|---|---|---|---|---|---|
| XP_016548138.1 | *CaCCR1-1* | PREDICTED: cinnamoyl-CoA reductase 1 | 975 | 324 | 35.8 | 5.36 | 11 |
| XP_016561561.1 | *CaCCR1-2* | PREDICTED: cinnamoyl-CoA reductase 1 | 1089 | 362 | 40.1 | 6.90 | 2 |
| XP_016562886.1 | *CaCCR1-like1* | PREDICTED: cinnamoyl-CoA reductase 1-like | 1005 | 334 | 36.9 | 6.41 | 3 |
| XP_016562887.1 | *CaCCR1-like2* | PREDICTED: cinnamoyl-CoA reductase 1-like | 1005 | 334 | 37.1 | 6.46 | 3 |
| XP_016576987.1 | *CaCCR1-like3* | PREDICTED: cinnamoyl-CoA reductase 1-like | 963 | 320 | 35.3 | 6.61 | 1 |
| XP_016577001.1 | *CaCCR1-like4* | PREDICTED: cinnamoyl-CoA reductase 1-like | 990 | 329 | 36.0 | 6.33 | 1 |
| XP_016558919.1 | *CaCCR1-like5* | PREDICTED: cinnamoyl-CoA reductase 1-like | 453 | 150 | 16.7 | 8.54 | 1 |
| XP_016556899.1 | *CaCCR1-like6* | PREDICTED: cinnamoyl-CoA reductase 1-like | 987 | 328 | 35.9 | 7.60 | 1 |
| XP_016538689.1 | *CaCCR1-like7* | PREDICTED: cinnamoyl-CoA reductase 1-like | 987 | 328 | 35.8 | 5.86 | 1 |
| XP_016562803.1 | *CaCCR1-like8* | PREDICTED: cinnamoyl-CoA reductase 1-like | 276 | 91 | 10.1 | 9.78 | 1 |
| XP_016564118.1 | *CaCCR1-like9* | PREDICTED: cinnamoyl-CoA reductase 1-like isoform X1 | 990 | 329 | 36.8 | 5.19 | 3 |
| XP_016575015.1 | *CaCCR1-like10A* | PREDICTED: cinnamoyl-CoA reductase 1-like isoform X1 | 945 | 314 | 35.3 | 4.97 | 6 |
| XP_016575016.1 | *CaCCR1-like10B* | PREDICTED: cinnamoyl-CoA reductase 1-like isoform X2 | 942 | 313 | 35.3 | 4.97 | 6 |
| XP_016538911.1 | *CaCCR2-1* | PREDICTED: cinnamoyl-CoA reductase 2 | 981 | 326 | 35.2 | 6.25 | 8 |
| XP_016578459.1 | *CaCCR2-2* | PREDICTED: cinnamoyl-CoA reductase 2 | 999 | 332 | 36.7 | 6.02 | 6 |
| XP_016568210.1 | *CaCCR2-like1* | PREDICTED: cinnamoyl-CoA reductase 2-like | 966 | 321 | 35.5 | 5.86 | 4 |
| XP_016572918.1 | *CaCCR2-like2* | PREDICTED: cinnamoyl-CoA reductase 2-like | 963 | 320 | 35.8 | 5.39 | 1 |

**Table 1.** *Cont.*

| ID | Name | Gene Description | ORF [b] | Protein Size [c] | Theoretical MW [c] (kDa) | pI | Chromosome |
|---|---|---|---|---|---|---|---|
| XP_016581443.1 | *CaCCR2-like3* | PREDICTED: cinnamoyl-CoA reductase 2-like | 1002 | 333 | 37.2 | 8.80 | 1 |
| XP_016564119.1 | *CaCCR2-like4* | PREDICTED: cinnamoyl-CoA reductase 2-like isoform X2 | 831 | 276 | 31.0 | 5.18 | 3 |
| NP_001311706.1 | *CaDFR-1* | PREDICTED: dihydroflavonol-4-reductase | 1149 | 382 | 42.5 | 562 | 2 |
| XP_016556013.1 | *CaDFR-like* | PREDICTED: dihydroflavonol-4-reductase-like | 1029 | 342 | 38.0 | 5.95 | Un |
| PHT59959.1 | *CbCCR1-1* | Cinnamoyl-CoA reductase 1 | 1002 | 333 | 37.2 | 8.46 | 1 |
| PHT59278.1 | *CbCCR1-2* | Cinnamoyl-CoA reductase 1 | 990 | 329 | 36.0 | 6.61 | 1 |
| PHT59279.1 | *CbCCR1-3* | Cinnamoyl-CoA reductase 1 | 963 | 320 | 35.3 | 6.61 | 1 |
| PHT46015.1 | *CbCCR2-1* | Cinnamoyl-CoA reductase 2 | 999 | 332 | 36.8 | 6.02 | 6 |
| PHT47585.1 | *CbCCR2-2* | Cinnamoyl-CoA reductase 2 | 1005 | 334 | 36.9 | 7.03 | 5 |
| PHT47586.1 | *CbCCR2-3* | Cinnamoyl-CoA reductase 2 | 1185 | 394 | 43.8 | 8.56 | 5 |
| PHT54849.1 | *CbDFR-1* | Cinnamoyl CoA reductase family/ Dihydroflavonol-4-reductase | 1356 | 451 | 50.3 | 8.49 | 2 |
| PHT56606.1 | *CbDFR-2* | Cinnamoyl CoA reductase family/ Dihydroflavonol-4-reductase | 1149 | 382 | 42.5 | 5.62 | 2 |
| PHU29451.1 | *CcCCR1-1* | Cinnamoyl-CoA reductase 1 | 963 | 320 | 35.3 | 6.61 | 1 |
| PHU29452.1 | *CcCCR1-2* | Cinnamoyl-CoA reductase 1 | 990 | 329 | 36.0 | 6.33 | 1 |
| PHU30004.1 | *CcCCR1-3* | Cinnamoyl-CoA reductase 1 | 1002 | 333 | 37.2 | 8.68 | 1 |
| PHU15306.1 | *CcCCR1-4* | Cinnamoyl-CoA reductase 1 | 1929 | 642 | 72.3 | 5.98 | 6 |
| PHU02385.1 | *CcCCR2-1* | Cinnamoyl-CoA reductase 2 | 1005 | 334 | 36.9 | 7.03 | 11 |
| PHU02386.1 | *CcCCR2-2* | Cinnamoyl-CoA reductase 2 | 1005 | 334 | 37.0 | 6.71 | 11 |
| PHU29448.1 | *CcDFR-1* | Cinnamoyl CoA reductase family/ Dihydroflavonol-4-reductase | 255 | 84 | 9.3 | 5.16 | 1 |
| PHU27058.1 | *CcDFR-2* | Cinnamoyl CoA reductase family/ Dihydroflavonol-4-reductase | 1149 | 382 | 42.5 | 5.45 | 2 |
| PHU07366.1 | *CcDFR-3* | Cinnamoyl CoA reductase family/ Dihydroflavonol-4-reductase | 192 | 63 | 6.9 | 9.03 | 9 |

*CaCCRs* were distributed across chromosomes 1, 2, 3, 4, 6, 8, and 11, and *CbCCRs* across chromosomes 1, 2, 5, and 6, *CcCCRs* across chromosomes 1, 2, 6, 9, and 11 (Table 1). In the *CA*, chromosome 1 included 8 *CaCCRs*, and chromosomes 6 and 3 contained 3 and 4 *CaCCRs*, respectively. Chromosomes 4, 8, 11, and 2 contained 1-2 *CaCCRs*. In the *CB* and *CC*, chromosome 1 included 3 *CbCCRs* and 4 *CcCCRs*, respectively, and the other chromosomes contain 1–2 *CCRs* (Table 1).

Intron–exon structures of the above *CCRs* and *CCR*-like genes showed that pepper *CCRs* were composed of 2–10 exons (Figure 1). Based on the number of exons and exon–intron structures, Barakat et al. [26] grouped *P. trichocarpa CCR* and *CCR-like* genes into three exon–intron patterns (Patterns 1–3), and Park et al., 2017, grouped *O. sativa CCR* and *CCR-like* genes into five exon–intron patterns (Patterns 2–6). Most previously studied functional *CCRs*, such as *AtCCR1*, *EuCCR* (*E. gunnii CCR*), *ZmCCR1* (*Z. mays CCR1*), *SbCCR1* (*Sorghum bicolor CCR1*), and *Os CCR1*, are composed of five exons, as in the exon–intron structure Pattern 2 [16,21,24,35]. In this study, 14 *CCRs* such as *CaCCR1-like1* had five exons, with a Pattern 2 exon–intron structure, and 13 *CCRs* such as *CcCCR1-2* had six exons as in Pattern 3. Although *CbCCR2-3* had six exons and *CcCCR1-4* had eight exons, their exon–intron structure was more similar to that of Pattern 2 and we term these Pattern 2-like. On the contrary, *CaDFR-like* and *CaCCR1-2* had five exons, their exon–intron structures were more similar to that of Pattern 3, so we term these Pattern 3-like (Figure 1). The four-exon structure of *CaCCR2-like4* is different to Pattern 1 in PoptrCCR1 and we designated this group as new Pattern 1. *CaCCR2-1* had three exons and *CbDFR-1* had 10 exons with a different pattern, so we named the two groups as Patterns 4 and 5, which are different from *O. sativa* [35]. The other pepper *CCRs* were composed of 2–4 exons with much shorter ORF lengths than those of functional CCR genes (Figure 1).

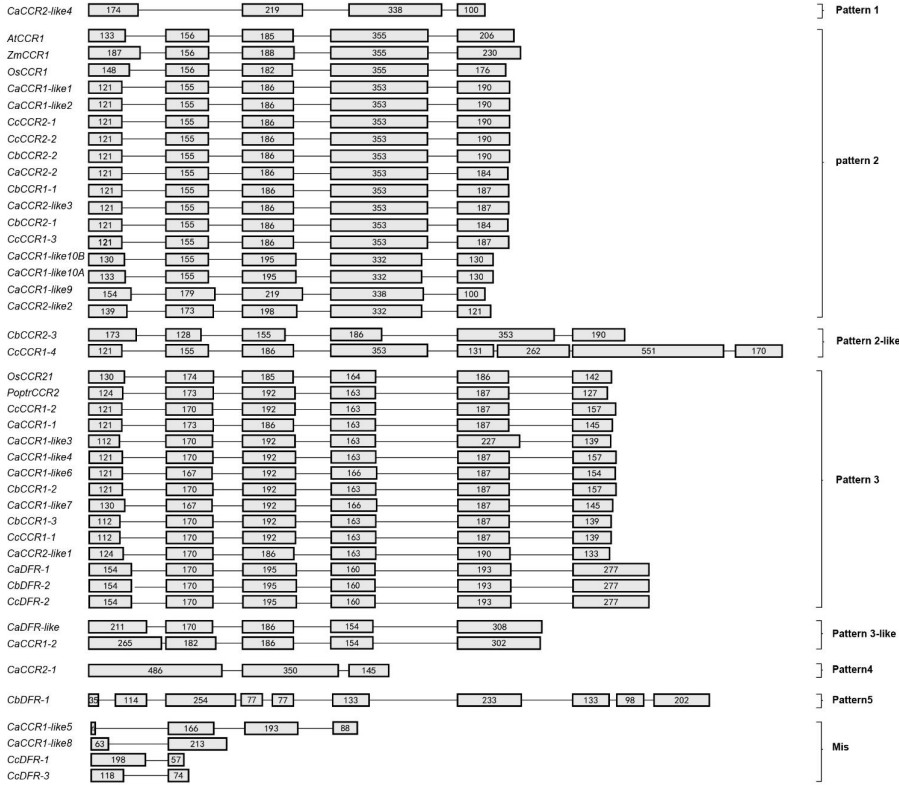

**Figure 1.** Exon–intron structures of pepper CCRs and other plant CCR genes. Pepper CCRs and other plant CCRs were divided into seven patterns based on their exon–intron structures. Exons and introns are indicated by boxes and lines, respectively. Numbers in the boxes represent the exon sizes. The intron sizes are not to scale. The pattern names of exon–intron structures are indicated in on the right side of the figure. *A. thaliana CCR (AtCCR)*, *Z. mays CCR (ZmCCR)*, *O. sativa CCR (OsCCR)*, and *Populus trichocarpa CCR (PoptrCCR)*.

### 3.2. Analysis of Cis-Regulatory Elements in Pepper CCR and CCR-like Gene Family

PLANTCARE software was used to analyze the sequence 2000 bp upstream from the initiation codon in pepper CCRs and revealed that the promoters of pepper CCRs contained four kinds of regulatory elements: growth and development, stress, hormone, and light response (Figure 2).

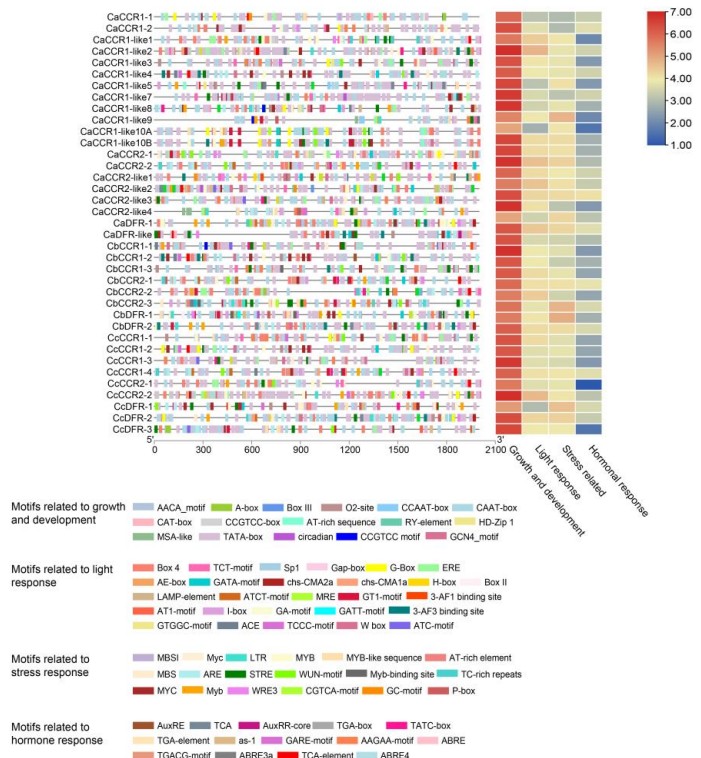

**Figure 2.** Analysis of cis acting elements of the promoter from the pepper CCR and CCR-like gene family.

All the promoters contained core regulatory elements CAAT-box and TATA-box in the growth and development elements. The AACA motif only exists in the promoter region of CcCCR1-2, box3 only exists in CaCCR2-like2, the RY element only exists in CbCCR 2–3, and the three elements appeared only once in their respective promoter. The most common element was O2 site, contained in the promoter region of 21 CCR and CCR-like genes (Table S3). A total of 27 types of 638 light response elements were found in the promoter of the CCR and CCR-like gene family, including 145 box4 distributed in 34 CCR and CCR-like gene promoters. There is only one H-box, which appears in the CaCCR2-like1 promoter. As the most light-response element in all promoters of families, CaCCR2-like2 and CcCCR2-2 contained 28 and 27 light response elements, respectively, mainly including box4, G-box, and ERE (Table S3).

We also uncovered a variety of abiotic-stress-responsive elements, such as drought-inducibility elements (MBS), low-temperature responsive elements (LTR), defense- and stress-responsive elements (TC-rich repeats, STRE), and wound-responsive elements (WUN-motif, WRE3). In total, 101 MYB response elements were included in the promoter of the *CCR* and *CCR-like* gene family except *CaCCR1-1* and *CaDFR-like*, which was the most responsive element in stress response. In contrast, only the *CcDRF3* promoter contained one MBS I element (flavonoid biosynthetic gene regulators) (Table S3). At the same time, we also found many plants hormone regulatory elements, such as methyl jasmonate (MeJA) responsive elements (TGACG-motif), abscisic acid responsive elements (ABRE, ABRE4 and ABR3a), auxin-responsive elements (TGA-element and AuxRR-core), gibberellin-responsive elements (GARE-motif), salicylic acid responsive elements (TCA-element), and Methyl jasmonate signal transduction responsive elements (TGACG-motif).

### 3.3. Sequence Homology and Phylogenetic Analysis of Pepper CCRs and CCR-like Protein Family

Multiple alignments revealed that pepper CCR protein sequences had about 4–95.5% similarity to functional CCRs from other plant species (Supplementary Table S4). In particular, CaCCR1-like1, CaCCR1-like2, CaCCR2-2, CaCCR2-like3, CbCCR1-1, CbCCR2-1, CbCCR2-2, CcCCR1-3, CcCCR2-1, and CcCCR2-2 had higher homologous with functional CCRs in other species. The short length CaCCR1-like5, CaCCTR1-like 8, CaCCR2-like4, CcDFR-1, CcDFR-3, and long length CbDFR-1 and CcCCR1-4 had a low sequence homology (<41.6%) similar to other plant CCRs (Supplementary Table S4). A phylogenetic analysis showed that 21 CaCCRs in *CA* and 26 CCRs derived from other species can be divided into gymnosperms, monocotyledons, dicotyledons, mixed Monocotyledons, and dicotyledons type. CaCCR1-like1, CaCCR1-like2, CaCCR2-2, and CaCCR2-like3 were grouped with SlCCR1, 2, and StCCR1 (Figure 3).

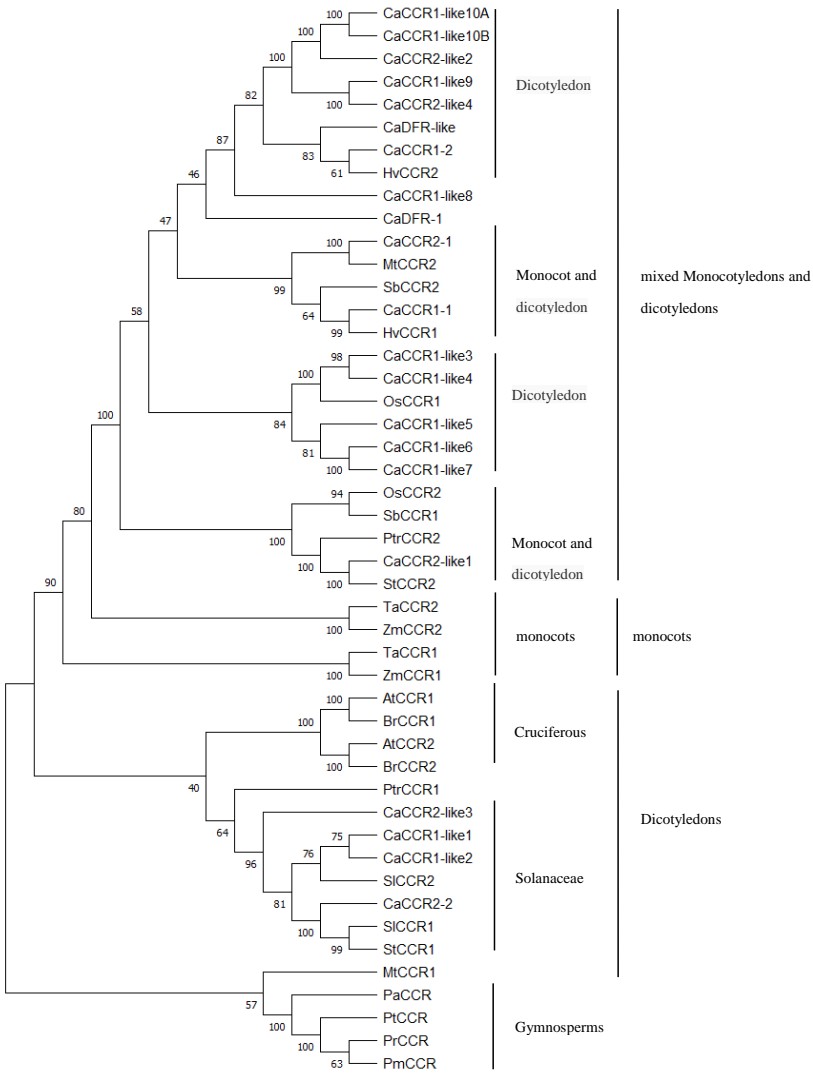

**Figure 3.** Phylogenetic tree analysis of CCR and CCR-like gene family protein sequences from *Capsicum annuum* L. and other species. *A. thaliana* CCRs (*AtCCR1*, AAG46037; *AtCCR2*, AAG53687); *Hordeum vulgare* (*HvCCR1*, XP_044948571.1; *HvCCR2*, KAE8782947.1); *Zea mays* (*ZmCCR1*, NP_001105488.1; *ZmCCR2*, NP_001105715); *Solanum tuberosum* (*StCCR1*, XP_006356249.1; *StCCR2*, XP_006342181.1); *Sorghum bicolor* (*SbCCR1*, XP_021306123.1; *SbCCR2*, XP_021312001.1); *T. aestivum* CCRs (*TaCCR1*, ABE01883; *TaCCR2*, AY771357); *Oryza sativa* (*OsCCR1*, XP_015621618.1; *OsCCR2*, XP_025883369.1); *Populus trichocarpa* (*PtrCCR1*, XP_006368516.2; *PtrCCR2*, XP_002300619.2); *Pinus massoniana* (*Pm*CCR, ACE76870.3); *Pinus taeda* (*Pt*CCR, AAL47684. 1); *Pinus radiata* (*Pr*CCR, AFC38436.1); *Picea abies* (*Pa*CCR, CAK18610.1).

The most striking homology between the predicted peptide sequences of pepper CCRs and functional plant CCRs was found in regions covered by two highly conserved motifs (Figure 4). These were the NAD(P)-binding motif at the N-terminus, and the catalytic motif for CCR activity. The former is a well-conserved motif for cofactor binding in the mammalian 3b-HSD/plant DFR protein superfamily [16,36,37]. The latter is a CCR signature motif (NWYCYGK), in which the NWYCY sequence may be crucial for CCR activity [16,26,37]. Especially, the NWYCY sequences of plant CCRs were identical in the blue dashed box (Figure 4). The above results suggest that CaCCR1-like1, CaCCR1-like2, CaCCR2-2, CaCCR2-like3, CbCCR1-1, CbCCR2-1, CbCCR2-2, CcCCR1-3, CcCCR2-1, and CcCCR2-2 were likely candidates for functional CCRs in pepper, with CaCCR1-like1, CaCCR1-like2, CaCCR2-2, and CaCCR2-like3 being the most plausible candidates.

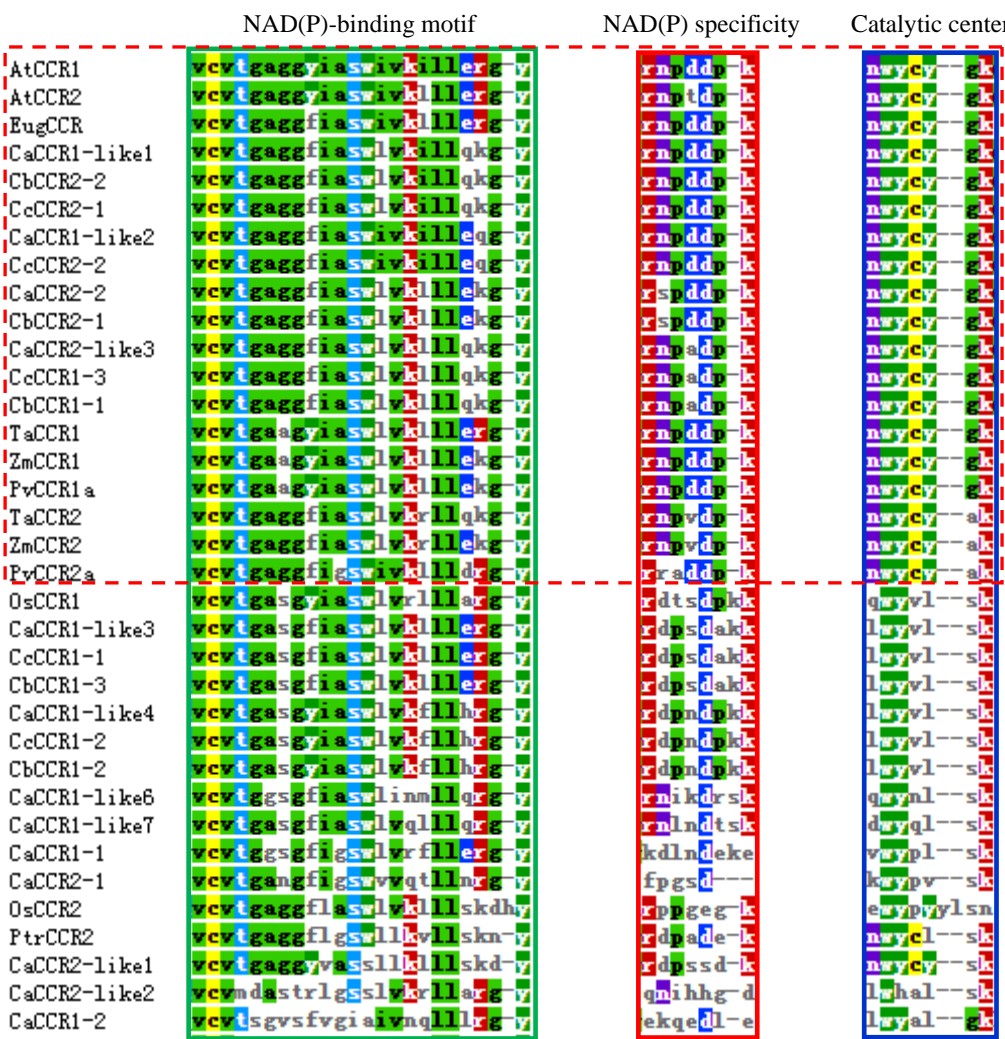

**Figure 4.** Multiple alignments of the NAD(P)-binding, NADP specificity, and CCR catalytic motifs of pepper CCRs with functional CCRs from other plant species. Amino acid sequences were aligned using Clustal-W. Colored amino acids denote identical or similar amino acids. NAD(P)-binding and NADP specificity motifs are boxed in green and red, respectively. The catalytic signature motif of CCRs is boxed in blue. *A. thaliana* CCRs (*AtCCR1*, AAG46037; *AtCCR2*, AAG53687); *O. sativa* (*Os-CCR1*, XP_015621618.1; *OsCCR2*, XP_025883369.1); *H. vulgare* CCR (*HvCCR*, AAN71760); *P. virgatum* CCRs (*PvCCR1a*, GQ450297; *PvCCR2a*, GQ450302); *T. aestivum* CCRs (*TaCCR1*, ABE01883; *TaCCR2*, AY771357); *Z. mays* CCRs (*ZmCCR1*, CAA74071; *ZmCCR2*, NP_001005715); *P. trichocarpa* (*PtCCR2*, XP_002300619.2); *E. gunnii* (*EugCCR*, CAA56103.1).

### 3.4. Cloning and Heterologous Expression of Two CCR Genes from Capsicum chinense

Based on the homology of CCR from different plant sources and the functional annotation of CCR in the CA genome, two full-length CCR cDNAs were cloned from Hainan huangdenglong pepper by PCR combined with RACE technology. Based on the conservative sites of the sequences in Figure 4, the BLAST search results, phylogenetic analysis homology between pepper CCRs and other plant CCRs (Figure 3 and Figure S4), the two cloned CCRs are named as CcCCR1 and CcCCR2, respectively. The *CcCCR1* cDNA is 1362 nucleotides long and has an open reading frame (ORF) of 1005 bp nucleotides encoding a 334 amino acid. The *CcCCR2* cDNA is 1423 nucleotides long and contains an ORF encoding a protein of 332 amino acids. No putative transmembrane domain and signal peptide are found in either *CcCCR*s. Amino acid sequence analysis showed that CcCCRs have a typical NAD (P) binding domain VCVTGAGGFIASWLVKILLQKGY in 9–31, and a conserved sequence (KNWYCYGK), in the amino acid sites of 144–151, considered as the catalytic site of CCRs.

Heterologous expressions of the His-tagged CcCCR1 and CcCCR2 proteins in *E. coli* by 0.1 mM IPTG and the recombinant protein (about 4kDa His tag) was obtained. The purified CcCCR1 and CcCCR2 proteins exhibited molecular masses of 35–40 kDa on SDS-PAGE, which agreed well with their theoretical molecular masses (Figure 5).

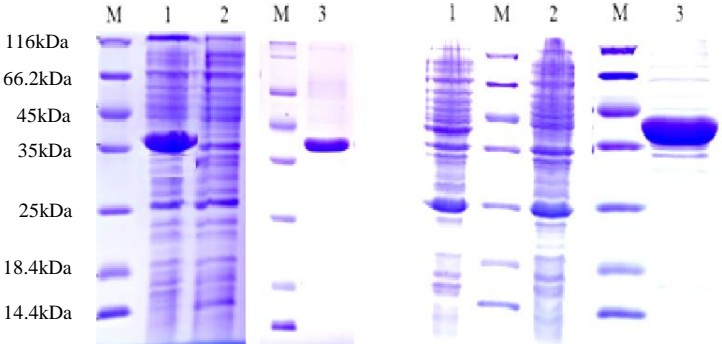

**Figure 5.** SDS-PAGE analysis of purified recombinant proteins from E. coli. M: molecular mass standards. Note: 5-Left: Lane 1: pET28a-CcCCR1; lane 2: empty pET28a vector; lane 3, purified pET28a-CcCCR1. 5-Right lane 1: pET28a-CcCCR2; lane 2: empty pET28a vector; lane 3, purified pET28a-CcCCR2.

### 3.5. CCR Activity and Kinetic Parameters of the Recombinant CcCCR1 and CcCCR2

To elucidate the enzymatic properties of cloned CcCCRs, the kinetic parameters of the recombinant CcCCR1 and CcCCR2 catalyzed reactions were determined toward the hydroxycinnamoyl-CoA substrate (Table 2). The Km-values of CcCCR1 for feruloyl-, sinapoyl-, and p-coumaroyl-CoA were 23.99, 32.35, and 34.52 μM, respectively (Table 2). The Kcat/Km values of CcCCR1 for feruloyl-CoA ($0.048 \ \mu M^{-1} \ min^{-1}$) was about two-fold higher than those for p-coumaroyl- and sinapoyl-CoAs (0.24 and $0.22 \ \mu M^{-1} \ min^{-1}$, respectively), indicating that it has a greater catalytic efficiency toward feruloyl-CoA than toward the other substrates (Table 2). The Km-values of CcCCR2 for feruloyl-, sinapoyl-, and p-coumaroyl-CoA were 16.53, 95.87, and 107.21 μM, respectively, indicating that CcCCR2 has a higher substrate affinity toward feruloyl-CoA than the other substrates (Table 2). The Kcat/Km values of CcCCR2 toward feruloyl-CoA ($0.819 \ \mu M^{-1} \ min^{-1}$) was about 50-fold and 8-fold higher than those for p-coumaroyl- or sinapoyl-CoAs (0.095 and $0.016 \ \mu M^{-1} \ min^{-1}$, respectively). This result indicates that among three hydroxycinnamoyl-CoA substrates, both CcCCR1 and 2 have substrate preferences for feruloyl-CoA, and CcCCR2 has a greater catalytic efficiency than CcCCR1.

**Table 2.** Kinetic parameters of CcCCR1 and CcCCR2.

| Eyzyme | Substrate | Km (µM) | Vmax (mmolL$^{-1}$min$^{-1}$) | Kcat (min$^{-1}$) | Kcat/Km (µM$^{-1}$min$^{-1}$) |
|---|---|---|---|---|---|
| CcCCR1 | Feruloyl-CoA | 23.99 ± 3.32 | 0.66 ± 0.10 | 1.14 ± 016 | 0.048 |
| | Sinapoyl-COA | 32.35 ± 5.96 | 0.45 ± 0.09 | 0.77 ± 0.16 | 0.024 |
| | *p*-Coumaroyl-CoA | 34.52 ± 3.69 | 0.78 ± 0.08 | 1.34 ± 0.14 | 0.022 |
| CcCCR2 | Feruloyl-CoA | 16.53 ± 1.09 | 9.20 ± 1.14 | 13.53 + 1.67 | 0.819 |
| | Sinapoyl-COA | 95.87 ± 9.58 | 1.06 ± 0.11 | 1.56 ± 0.16 | 0.016 |
| | *p*-Coumaroyl-CoA | 107.21 ± 3.13 | 6.92 ± 1.21 | 10.17 ± 1.78 | 0.095 |

The recombinant CcCCR1 maintained high catalytic activity between the reaction system pH 6.0–6.5. When the pH was more than 6.5, the enzyme activity decreased rapidly, and when the reaction system temperature was 30–40°C, the enzyme showed good activity, and the enzyme was almost inactivated at the highest temperature of 80 °C and the lowest temperature of 10 °C. The activity of recombinant CcCCR2 was the highest at pH 5.5, and then decreased rapidly, while the protein maintained high catalytic activity at 40 °C (Figure 6).

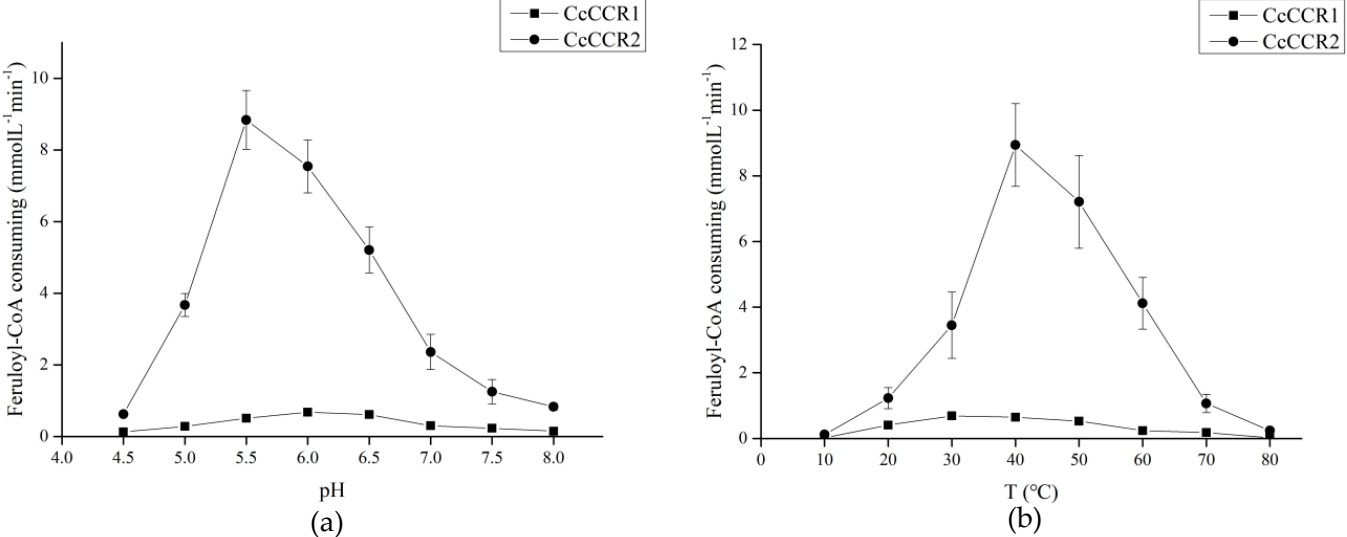

**Figure 6.** The pH (**a**) and temperature (**b**) profiles for recombinant CcCCR1 and CcCCR2 activities.

*3.6. The Correlation between Expression of CcCCRs and Contents of Capsaicin, Lignin, and Flavonoids*

The capsaicin content of the three pepper varieties was different due to different parts and different fruit development stages (Figure 7a–c). The capsaicin content of Huangdeng-long pepper was very significantly higher than that of the other two varieties, and the spicy flavor cannot be detected in Zhongtian F1, a bell pepper. In the same variety, the hottest part appeared in placental tissue, followed by pericarp, and the spicy flavor could not be detected in stems and leaves (Figure 7a–c). In addition, the capsaicin content increased first and then decreased with the number of fruit development days, and the highest peak appeared 40 days after flowering. At this time, the capsaicin content in placental tissue of Huangdenglong pepper reached 6.22 mg/g (Figure 7a,b).

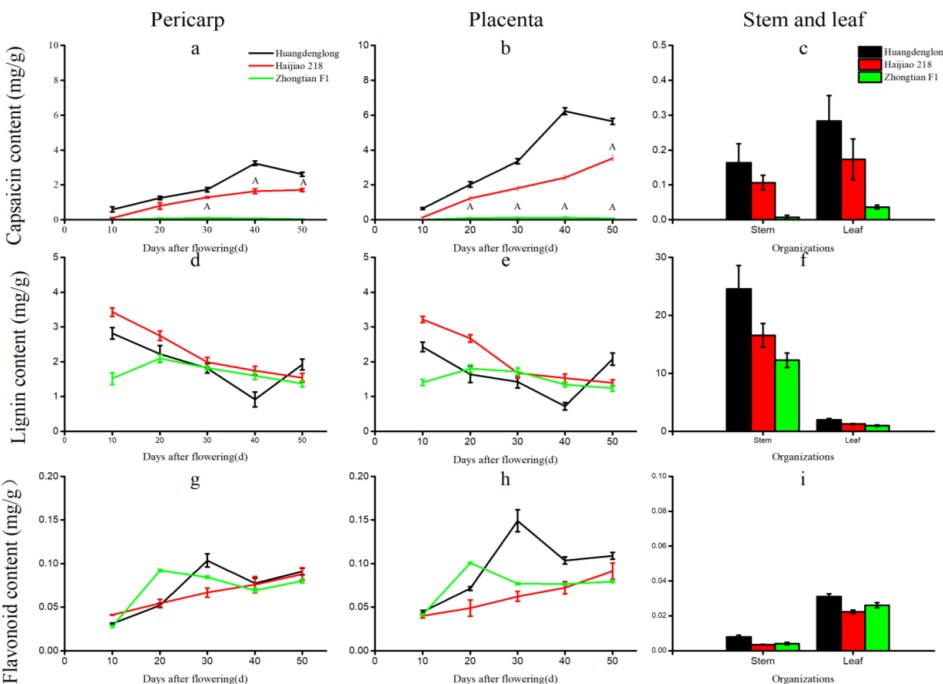

**Figure 7.** Comparison of capsaicin, lignin, and flavonoids in different tissues at different stages. Note: (**a**,**b**,**d**,**e**,**g**,**h**) The abscissa is the days after flowering, and the ordinate is the content of capsaicin/lignin/flavonoids in pericarp and placenta of different peppers. (**c**,**f**,**i**) The abscissa is different organizations, and the ordinate is the content of capsaicin/lignin/flavonoids in organizations of different pepper varieties.

Similarly, lignin content varied with different pepper varieties, parts, placentation, and pericarp development stages (Figure 7d–f). The lignin content in the stems (24.58 mg/g) and leaves (2.00 mg/g) of the Huangdenglong pepper were higher than that of the other two varieties (Figure 7f). In general, the lignin content of the pericarp and the placenta of three varieties decreased with the number of fruit development days. Among them, the Huangdenglong pepper decreased to the lowest point at 40 days (0.92 mg/g in pericarp and 0.72 mg/g in placenta, respectively), and then increased, and the varieties Haijiao 218 decreased (from 3.43 to 1.54 mg/g in pericarp and 3.20 to 1.40 mg/g in placenta), and Zhongtian F1 showed a slight increase before 20d, then slowly decreased (Figure 7d,e).

The content of flavonoids in pericarp and placenta of three pepper varieties showed an upward trend with the number of fruit development days, which was consistent with the change trend of capsaicin content, and the content in leaves was higher than that in stems (Figure 7g–i). However, the content of flavonoids in pepper is generally low, and the content of flavonoids in all parts of the three varieties ranges from 0.08 to 0.15 mg/g (Figure 7g–i).

In order to analyze the relationship between CCR expression level and spicy flavor, we further measured the CCR expression in different parts and fruit development stages of Huangdenglong pepper. The results showed that *CcCCR1* and *CcCCR2* expression levels in Huangdenglong stem and leaf were significantly higher than that in pericarp and placenta (Figure 8). Correlation analysis showed that the capsaicin content was extremely positively correlated with flavonoids (R = 0.746, $p$ = 0.01), extremely negatively correlated with CcCCR2 relative expression level (R = −0.664, $p$ = 0.01), significantly negatively correlated with CcCCR1 relative expression level ($r$ = −0.414, $p$ = 0.05) and lignin content ($r$ = −0.387, $p$ = 0.050) in Huangdenglong pepper. The lignin content significantly positively correlated with CcCCR2 expression level ($r$ = 0.326, $p$ = 0.05), but did not correlate with CcCCR1. These results showed that CCR2 had a greater negative effect on pungency formation (Table 3).

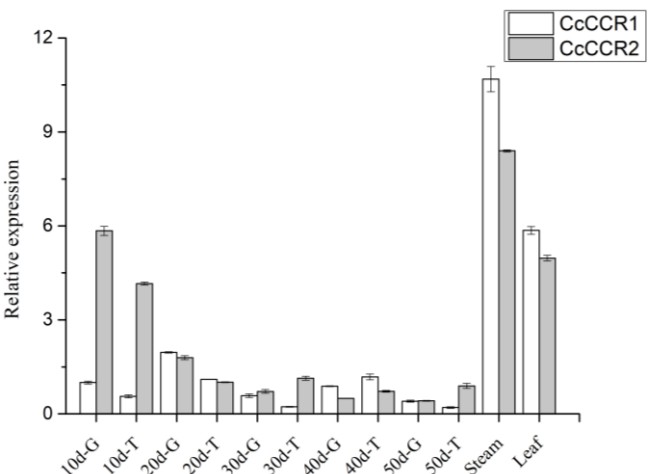

**Figure 8.** Relative expression of CcCCRs. G means pericarp; T means placenta.

**Table 3.** Correlation coefficients between CcCCRs relative expression level and metabolite content in Huangdenglong.

| Factor | Capsaicin | Lignin | Flavonoid | *CcCCR1* | *CcCCR2* |
|---|---|---|---|---|---|
| Capsaicin | 1 | | | | |
| Lignin | −0.387 * | 1 | | | |
| Flavonoid | 0.746 ** | −0.534 ** | 1 | | |
| *CcCCR1* | −0.414 * | 0.267 | −0.252 | 1 | |
| *CcCCR2* | −0.664 ** | 0.326 * | −0.719 ** | 0.590 ** | 1 |

Note: 1 * the correlation coefficient is significant at 0.05 level. 2 ** The correlation coefficient is significant at 0.01 level.

### 3.7. Functional Confirmation in the Regulation of Pungency in Pepper

The expression vectors pBI121-*CCRs* containing *CcCCR1* and *CcCCR2* genes were constructed to study the function of CcCCRs, respectively, and the *CcCCR1* and *CcCCR2* genes were successfully transformed into *Arabidopsis* by Agrobacterium-mediated transformation. The lignin content of transformed *Arabidopsis* T2 plants with *CcCCR 1* and *CcCCR 2* increased by 13.02 and 24.46% on average, and the maximum lignin content per plant increased by 14.07 and 27.01%, respectively (Figure 9). This reveals that the expression of *CcCCR1* and *CcCCR2* can affect the content of lignin.

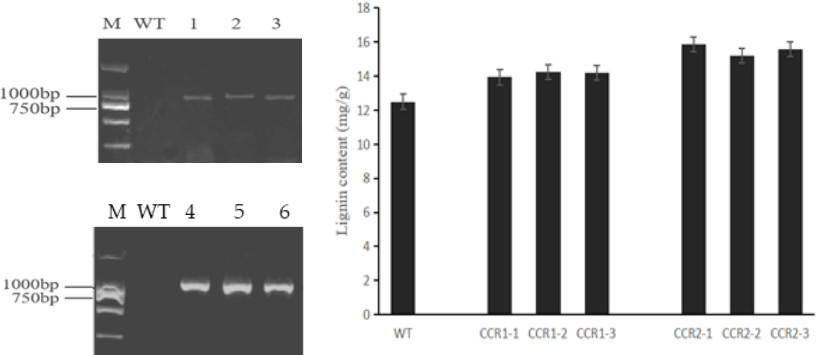

**Figure 9.** PCR and lignin content detection of *Arabidopsis thaliana*. Note: CCR1-1, CCR1-2, and CCR1-3 were three different CCR1 transgenic Arabidopsis plants. CCR2-1, CCR2-2, and CCR2-3 were three different CCR2 transgenic Arabidopsis plants. M: DL2000 marker; 1–3: CcCCR1 transgenic plants, 4–6: CcCCR2 transgenic plants.

In order to confirm the role of CCR in the regulation of spicy taste of Huangdenglong pepper, pTRV1, pTRV2, and pTRV2-CcCCRs silencing vectors containing *CCR1* and *CCR2*

were introduced into *Agrobacterium tumefaciens* (Figure 10a), and the fruit stalk of young fruit of about 5 DAF was injected by injection method in Huangdenglong pepper (Figure S3). Two weeks after inoculation, the relative expression level of *CcCCR1* in pericarp, placenta, and fruit stalk of *CcCCR1* silent plants were 6, 6.4, and 12.9% of the control, respectively (Figure 10b), with the decrease of 62, 48.7, and 54.2% for *CcCCR2* and 83, 82.1, and 82.7% for CAD, respectively (Figure 10c,d). On the contrary, the relative expression level of *PAL*, *pAMT*, and *CS* genes in placenta increased by 1.7, 2.26, and 0.38 times, respectively. The relative expression level of *4CL* in pericarp and placenta did not change much (Figure 10e–h).

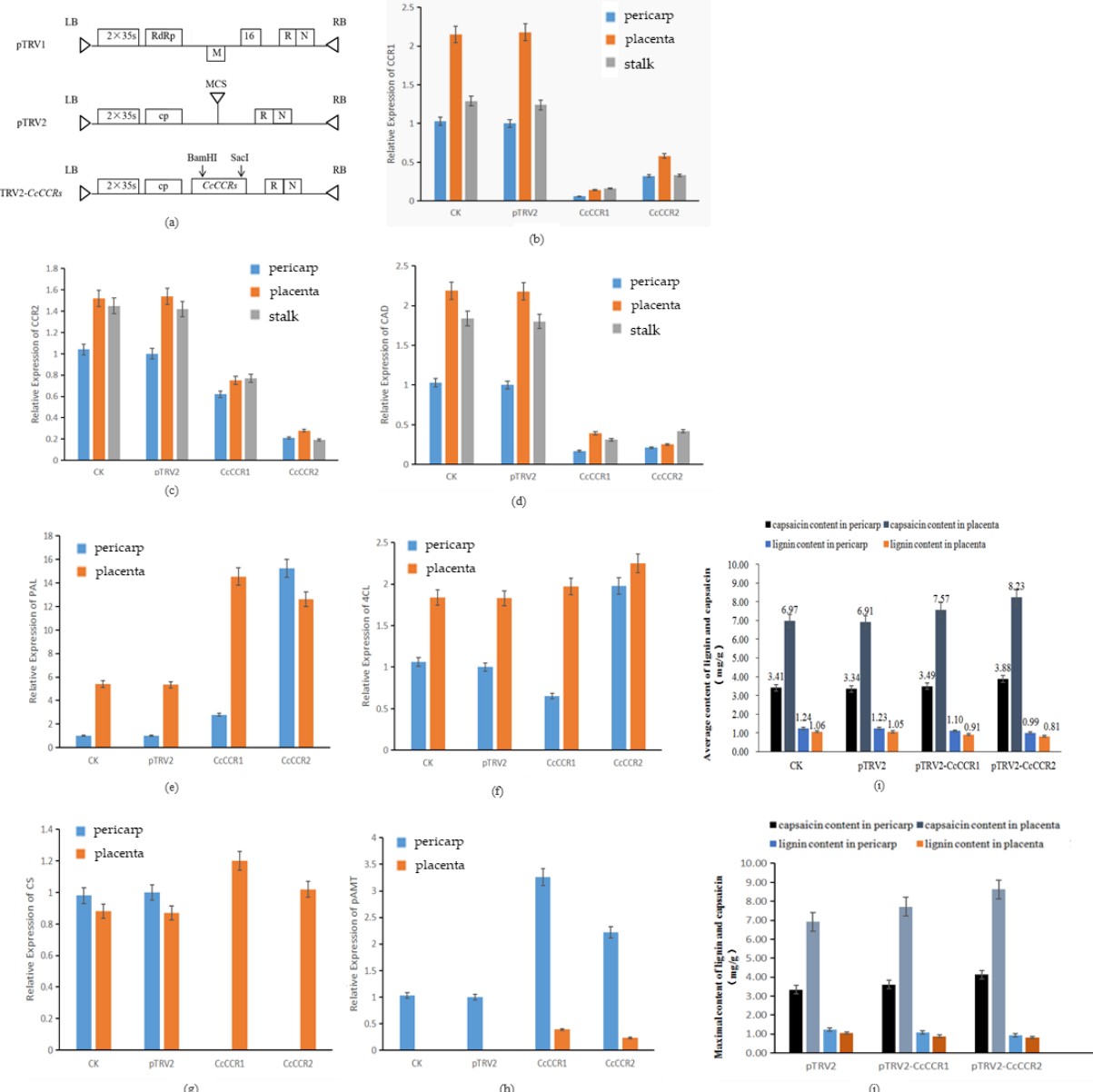

**Figure 10.** Determination of candidate gene expression and capsaicin and lignin content after inoculation. Note: (**a**) Vector construction, CCR gene insertion site. (**b–h**) The expression level of different genes under different treatments. (**i,j**) Contents of capsaicin and lignin in placenta and pericarp under different treatments. CK: blank control; pTRV2: pTRV2 no-load plant; CcCCR1/CcCCR2: silenced gene plant.

In the *CcCCR2* silenced plants, the expression level of *CcCCR* in pericarp, placenta, and fruit stalk were 21%, 18.2%, and 13.4% of the corresponding tissues of the transformed control, respectively, with the decrease of 68% to 88.5% for CcCCR1 and CAD (Figure 10b–d).

In pericarp and placenta of pepper fruits, the expression of *pal*, *4CL,* and *pAMT* increased, and the expression of *pal* in pericarp increased more than 15 times. The expression of CS in placenta increased by 1.17 times, but it was not detected in pericarp (Figure 10e–h).

Further detection of lignin and capsaicin contents in pericarp and placentation tissue of 40 days after flowering in silenced, transformed empty vector, and non-transformed Huangdenglong pepper plants showed that the lignin content decreased by 10.2 and 13.5% in the pericarp and placenta of *CcCCR1* silenced fruit, respectively, while the capsaicin content increased by 5 and 9.6%, respectively (Figure 10i). In *CcCCR2* silenced fruit, the lignin content in pericarp and placenta decreased by 18.8 and 22.8%, respectively; however, the capsaicin content increased by 16.2 and 19.1%, respectively (Figure 10i). In particular, the lignin content decreased by 24.0 and 23.3% in fruit pericarp and placental tissue of *CcCCR2* silent lines with the largest change, accompanied by the largest increase in capsaicin content by 23.5 and 24.9% (Figure 10j). These results indicated that the *CcCCR2* gene plays a more important role than CcCCR1 in decreasing the lignin content and increasing the capsaicin content. It is speculated that the CcCCRs gene in Huangdenglong pepper positively regulates lignin biosynthesis.

The content of capsaicin and lignin in pericarp and placenta of 40 DAA (days after anthesis) showed that the content of capsaicin and lignin in pTRV2 was slightly lower than that in CK plants, but there was no significant difference between them. The average capsaicin content in the pericarp of *CcCCR1* silenced plants was 3.49 mg/g, which was not significantly different from 3.34 mg/g of pTRV2 empty. The average capsaicin content in placenta was 7.57 mg/g, which was 9.6% higher than 6.91 mg/g of pTRV2 empty. Accordingly, the lignin content in the pericarp of *CcCCR1* silenced plants was 1.1 mg/g, which was 10.2% lower than the no-load content, and the lignin content in the placenta was 0.907 mg/g, which was 13.5% lower than the no-load content. The lignin content corresponding to the maximum capsaicin content of a single pepper was reduced by 12 and 16.6% in the pericarp and placenta, respectively.

## 4. Discussion

With the completion of genome sequencing of many species, screening and identification of the mammalian 3b-hydroxysteroid dehydrogenase (HSD)/plant dihydroflavonol reductase (DFR) superfamily of lignin biosynthesis-related members have been performed in *P. trichocarpa*, *O. sativa*, *Lolium perenne*, *Pyrus bretschneideri,* and other plants [38–42]. However, there is no such systematic study of CCR families in pepper. In this study, 21 predicted CCRs and CCR-like members were identified in the genome of *Capsicum annuun*, with 8 in *C. baccatum* and 9 in *C. chinense*, respectively. More CaCCRs were found in pepper than in Arabidopsis (11), *L. perenne* (13), *P. trichocarpa* (11), but fewer than in *Eucalyptus grandis* (27), *P. bretschneideri* (31), and *Malus pumila* (47). Consistent with most reported plant CCRs, the isoelectric point and protein molecular weight of pepper CCRs and CCR-like range from 4.97 to 7.03 and 35.2 to 37.2 KDa, respectively. Exon–intron structure analysis of pepper CCRs and other plant CCR genes indicated that CaCCR1-like1, CaCCR1-like2, CaCCR2-2, CaCCR2-like3, CbCCR1-1, CbCCR2-1, CbCCR2-2, CcCCR1-3, CcCCR2-1, and CcCCR2-2 are composed of five exons, which is similar to the exon–intron structure Pattern 2 of *AtCCR1*, *EuCCR* (*E. gunnii CCR*), *ZmCCR1* (*Z. mays CCR1*), *SbCCR1* (*Sorghum bicolor CCR1*), and *Os CCR1* [16,21,24,35].

Our results also showed that the amino acid sequences of CaCCR1-like1, CaCCR1-like2, CaCCR2-2, CaCCR2-like3, CbCCR1-1, CbCCR2-1, CbCCR2-2, CcCCR1-3, CcCCR2-1, and CcCCR2-2 had 58.8–66.7% identity with ZmCCR1/2, and 68.7–75.9% with AtCCR1/2 (Supplementary Table S4), which is similar to the previous research results that ZmCCR1, 2 and EuCCR were 74 and 66%, respectively [33]. This result indicated that CCR proteins are highly conservative in monocotyledons and dicotyledons (Supplementary Table S4). Based on functional differentiation, plant CCR can usually be divided into a bona fide CCR clade and a CCR-like clade, and AtCCR1/2 and ZmCCR1/2 belong to a bona fide CCR clade [41]. CaCCR1-like1, CaCCR1-like2, CaCCR2-2, CaCCR2-like3, and AtCCR1/2 are not only

closely related, but also belong to dicotyledons in the phylogenetic analysis (Figure 3), and the motif consensus (NWYCY) without mismatch was considered to be the symbol of real CCR [26]; therefore, it is reasonable to believe that CaCCR1-like1, CaCCR1-like2, CaCCR2-2, and CaCCR2-like3 belong to a bona fide CCR clade. In previous studies, *AtCCR*, *PhCCR*, *HvCCR1*, *PdCCR*, and *StCCR* all played an important role in lignin synthesis [20,43–47], so it can be speculated that pepper CCRs also has the function of regulating lignin.

Based on the high homology with CaCCR1-like1, CaCCR1-like2, CaCCR2-2, CaCCR2-like3, we cloned CcCCR1 and CcCCR2 from Huangdenglong pepper and successfully expressed in *E. coli*. Evolutionary relationship analysis showed that CcCCR1 had high homology with CaCCR1-like1 and CaCCR1-like2, but CcCCR2, CaCCR2-2, SlCCR1, and StCCR1 clustered together and were closer to AtCCR1/2, CaCCR2-like3, respectively. The CCR enzyme catalyzing the first step in the monolignol-specific branch of the lignin biosynthetic pathway used four cinnamoyl CoA esters (p-coumaryl-CoA, caffeoyl-CoA, feruloyl-CoA, 5-hydroxyferuloyl-CoA, and sinapoyl-CoA) as substrates and some *CCR* genes have preference for certain substrates. For instance, TaCCR1, AtCCR1/2, EgCCR [30], Ph-CCR1, PvCCR1 [19], MtCCR1 Ll-CCRH1 [48], and our cloned CcCCR1/2 (Table 2) preferentially use feruloyl-CoA compared to sinapoyl-CoA and p-coumaryl-CoA, and had a higher conversion rate for feruloyl-CoA. However, MtCCR2 and PvCCR2 prefer caffeoyl- and p-coumaroyl- CoAs, and TaCCR2 used feruloyl CoA, 5-OH-feruloyl CoA, sinapoyl CoA, and caffeoyl CoA with equal efficiency [27]. In addition, although the acting substrate is the same, the catalytic efficiency of different CCRs sometimes varies greatly. AtCCR2 affinity and conversion efficiency of feruloyl-CoA and sinapoyl-CoA were approximately five times lower compared to AtCCR. In our results, the conversion efficiency of feruloyl-CoA catalyzed by CcCCR2 is about 20 times that of CcCCR1 (Table 3).

Expression pattern and functional relationship analysis showed that PvCCR1 was probably associated with lignin biosynthesis during plant development, whereas PvCCR2 may function in defense [19], and Mt-CCR1 was the major CCR involved in lignin biosynthesis, whereas Mt-CCR2 was proposed to be involved in an alternative route for S lignin biosynthesis in *M. truncatula* [27]. In our results, the *CcCCR2* expression level has a significant correlation with lignin content, so it may play a more key role in regulating lignin synthesis than CCR1 (Table 3).

Downregulation of the CCR genes in a variety of species has been shown to redirect metabolic flux away from developmentally related lignification and enhances the levels and composition of some of the stress-related phenylpropanoid intermediates and derivatives in CCR-deficient plants. Silencing of the CCR gene in tobacco results in decreased flux from feruloyl-CoA to G and S units, respectively, leading to the observed reduction in the level of lignin-specific phenolic molecules [49]. RNAi of the *CCR* gene decreased the lignin content in tomato and increased the soluble phenol content in stems and leaves of transformed plants [50]. CCR downregulation of poplars resulted in an orange to wine-red coloration of the xylem that often appears in patches along the stem, which is associated with a reduction in lignin amount and the incorporation of low levels of ferulic acid into the polymer [18], but is helpful for increasing ethanol yield of wood [25]. Mu insertional mutant showed that *CCR1* gene expression is reduced to 31% of the residual wild-type level, which leads to a slight decrease in lignin content in $ZmCCR1^-$ and upregulation of several flavonoid biosynthetic genes including a chorismate mutase, two dihydroflavonol reductases (DFR), and a flavonoid 3′-hydroxylase (F3H), indicating that phenylpropanoid intermediates are redirected towards the flavonoid pathway [24]. Although there are few studies on CCR overexpression, studies in *Brassica napus* showed that BnCCR1 showed stronger effects on lignification-related development, phenylpropanoid flux control, and seed coat pigmentation, whereas BnCCR2 showed a stronger effect on sinapate biosynthesis, and their effects on glucosinolate metabolism are almost opposite [42]. We not only realized the heterologous expression of CCR in Arabidopsis, increased the lignin content of Arabidopsis, but also successfully carried out VIGS in pepper, reduced the lignin content of pepper fruit, and increased the capsaicin content. At the same time, we also found that CCR silencing

decreased the CAD expression of the phenylpropane metabolic lignin synthesis pathway gene, but increased the levels of *PAL*, *PAMT*, and *CS*, which are the enzymes of capsaicin biosynthesis [51,52].

## 5. Conclusions

In this paper, 38 genes were found as predicted CCRs or CCRs-like (CCR/DFR/epimerase 3b-HSD) proteins in different pepper genomes, respectively, and divided into six patterns according to exon–intron structure models. The promoters of pepper CCRs were predicted to contain growth and development, stress, hormone, and light response elements. Two confidential CCR genes named CcCCR1 and CcCCR2 were cloned from Capsicum chinense Jacquin. The results of enzyme characteristics showed that the affinity and catalytic efficiency of CcCCR1/2 to feruolyl-CoA was the highest. The results of heterologous expression, correlation analysis, ectopic expression, and VIGS showed that the highest expression of *CcCCR1* was found in stems, followed by leaves, and the higher expression of *CcCCR2* was found in stem and early fruit than other organs. Capsaicin content was negatively correlated with lignin and positively correlated with flavonoids. CcCCR2 expression level has a significant correlation with lignin content, so it may play a more key role in regulating lignin synthesis than CcCCR1. CCR2 played a more important role in regulating pungency formation.

**Supplementary Materials:** The following supporting information can be downloaded at: https://www.mdpi.com/article/10.3390/horticulturae8060537/s1, Figure S1: Proposed model of the capsaicin biosynthetic pathway [5–8]; Figure S2: Phenylpropanoid metabolic pathway and possible intermediate products in plants [9,10]; Figure S3: Inoculation period and site; Figure S4: Phylogenetic analysis homology between pepper CCRs and other plant CCRs; Table S1: Primers for cloning and expression; Table S2: Primers for vigs; Table S3: Promoter element; Table S4: Sequence similarity.

**Author Contributions:** Conceptualization, D.W., M.N. and X.L.; methodology, D.W., M.N., X.L., W.Z. and L.Z.; software, D.W., M.N. and X.L.; validation, D.W., M.N., X.L., W.Z. and L.Z.; formal analysis, D.W., M.N. and X.L.; investigation, D.W., M.N., X.L., W.Z., H.S., and L.Z.; resources, S.C.; data curation, W.Z., L.Z. and H.S.; writing—original draft preparation, S.C., D.W., M.N. and X.L.; writing—review and editing, S.C., Z.W., J.Z., P.L., H.L. and L.Y.; supervision, S.C., Z.W., J.Z., P.L., H.L. and L.Y.; project administration, S.C.; funding acquisition, S.C. All authors have read and agreed to the published version of the manuscript.

**Funding:** This research was funded by the National Natural Science Foundation of China (Grant no. 31760578), and National Natural Science Foundation of China (Grant no. 2019RC030).

**Institutional Review Board Statement:** Not applicable.

**Informed Consent Statement:** Not applicable.

**Data Availability Statement:** All relevant data is shown in figures.

**Acknowledgments:** Thanks to all teachers and students for their cooperation and help.

**Conflicts of Interest:** The authors declare no conflict of interest.

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
