# Peer review of "Analyses of Pepper Cinnamoyl-CoA Reductase Gene Family and Cloning of CcCCR1/2 and Their Function Identification in the Formation of Pungency"

_horticulturae, doi:10.3390/horticulturae8060537_

Round 1

Reviewer 1 Report

Please find document attached

Author Response

Introduction

  • Line 51, please correct “accounts” for 69% of the total capsaicinoid

It has been revised.

  • Line 93, does not reads well and needs revision “but CCR2 prefers caffeoyl and 4-coumaroyl CoAs, exhibits sigmoidal kinetics with these substrates [26,27].

It has been revised---“CCR1 exhibits preference for feruloyl CoA, while CCR2 prefers caffeoyl and 4-coumaroyl CoAs, and both of them exhibit sigmoidal kinetics with these substrates”

  • Line 100-101, could be improved

It has been revised.

  • Line 106, please correct typo “was conformed to estabolish the competitive relation”

It has been revised.

Material & Methods

  • Plant material also include the reagents? Sigma-Aldrich origin description?

It has been revised. Sigma-Aldrich (St. Louis, MO, USA).

  • Line 130: MEGA 6 using the neighbor-joining method?

It has been revised---“In MEGA 6 software, neighbor-joining (NJ) method was used to make phylogenetic analysis.”

  • Line 135: what parameters were used?

No additional parameters need to be set when using Plantcare to predict promoter cis-elements online.

  • Subsection 2.1 would need more details of the methods and parameters used.

It has been revised.

  • Subsection 2.5. I would encourage to add more details of the equipment and parameters used to perform the essays. For example, which UV spectrophotometer? or SDS-PAGE model? wich RT-PCR? etc.

The brand model has been added.

  • Line 196: Is this line correct? “According to the full-length sequences of CcCCR1 and CcCCR1 genes…”

It has been revised.

  • Subsection 2.8: RT-PCR?

Quantitative real-time PCR

Results:

  • It would help to the reader to clarify at the beginning within the text the differences among CaCCRs, CbCCRs or CcCCRs, line 523 is to far away.

It has been revised.

  • Line535: Please revise this line “had the identity of 58.8% to 66.7% with ZmCCR1/2, and 68.7% to 75.9% ….

It has been revised---“Our results also showed that the amino acid sequences of CaCCR1-like1, CaCCR1-like2, CaCCR2-2, CaCCR2-like3, CbCCR1-1, CbCCR2-1, CbCCR2-2, CcCCR1-3, CcCCR2-1 and CcCCR2-2 were 58.8% to 66.7% identity with ZmCCR1/2, and 68.7% to 75.9% with AtCCR1/2 (Supplementary Table 4), which is similar to the previous research results that ZmCCR1, 2 and EuCCR were 74% and 66% respectively[33]. This result indicated that CCR proteins are highly conservative in monocotyledons and dicotyledons.”

  • I would encourage to revisit the Results section. I believe the material presented is of high quality but I highly encourage streamline the text and present in a more succinct manner.

It has been revised.

Conclusions:

  • The conclusion needs revision, for example, the first paragraph. “The results obtained in this first study of analyze genome-wide CCR and CCR-like gene families and their exon-intron structures, sequence homology, phylogenetic characteristics and promoters”

It has been revised.

Reviewer 2 Report

This manuscript by Dan et al. analyzed the Cinnamoyl-CoA reductases gene families from peppers. The CCR gene structures, sequence homology, phylogenetic characterization and promoters were analyzed in three pepper species and also compared with some studied CCRs in other plants. They further characterized two CCR genes from Capsicum chinense. They found these two genes can affect the capsaicin content in peppers. Here are some concerns I want to point out.

Main points

  1. The main question is no statistical analysis was performed in many of the results. For example, in Figure 7, Figure 8, and Figure 10, there are many bar charts to indicate either the gene expression level or the content difference of capsaicin, lignin, and flavonoid in different samples. The statically analysis should be done to confirm whether these changes are significant. Also, in line 447 “…Huangdenglong stem and leaf were significantly higher than that in pericarp and placenta.” The word “significantly” means after statistical analysis, but the authors didn’t set any statistical analysis. 
  2. In section 3.4, the authors chose two CCR genes to study. But the rationale for selecting these two genes is not very clear. There are 9 CcCCRs in Capsicum chinense. Whythe authors chose these two is not very well explained. 

Minor points

  1. In line 22, the abbreviation of “Cinnamoyl-CoA reductases” should be added after the full name when used for the first time.
  2. In line 86, references should be added after the sentence “9 in Populus trichocarpa, 33 in Oryza sativa and 10 in Eucalyptus grandis”.
  3. In line 454, “These results showed that CCR2 had a greater effect on pungency formation”. Please make it clear about what kind of effect CCR2 has.
  4. The format of references is not consistent; some of them use the full name of journals, while the others are abbreviated names. 

 Author Response

Main points

1.The main question is no statistical analysis was performed in many of the results. For example, in Figure 7, Figure 8, and Figure 10, there are many bar charts to indicate either the gene expression level or the content difference of capsaicin, lignin, and flavonoid in different samples. The statically analysis should be done to confirm whether these changes are significant. Also, in line 447 “…Huangdenglong stem and leaf were significantly higher than that in pericarp and placenta.” The word “significantly” means after statistical analysis, but the authors didn’t set any statistical analysis.

Univariate comparisons were analyzed for significance. There is no comparison between multiple factors, so significance is not shown in the figure.

2.In section 3.4, the authors chose two CCR genes to study. But the rationale for selecting these two genes is not very clear. There are 9 CcCCRs in Capsicum chinense. Why the authors chose these two is not very well explained.

CcCCR1 and CcCCR2 cloned in this paper were homologous clones of CaccR1-like1 and CaCCR2-2 as reference genes, because there was no C. chinense genomic information before. Bioinformatics analysis of CCR was performed after the C.chinense genome was measured.

Minor points

1.In line 22, the abbreviation of “Cinnamoyl-CoA reductases” should be added after the full name when used for the first time.

It has been added.

2.In line 86, references should be added after the sentence “9 in Populus trichocarpa, 33 in Oryza sativa and 10 in Eucalyptus grandis”.

They have been added.

3.In line 454, “These results showed that CCR2 had a greater effect on pungency formation”. Please make it clear about what kind of effect CCR2 has.

It has been revised.

4.The format of references is not consistent; some of them use the full name of journals, while the others are abbreviated names.

All have been changed to abbreviation

Round 2

Reviewer 1 Report

Dear Authors, Editor,

The manuscript has been improved. Minor but important editing to consider are:

-Tables and figures, even though it may be common knowledge for some, should be self-explanatory. Please provide more details on Figure 2, e.g. color scale, acronyms, etc. and Table 4, e.g. acronyms, etc. , and definitively in the captions of Figure 7, Figure 9 and Figure 10. e.g. acronyms, classification for each sub graph, relevant details, etc.

Best wishes

Author Response

The full names in Figure 2 and Figure 4 are shown in Table 1. The color of the response element in Figure 2 is randomly generated by the software, and the width is subject to the length of the response element sequence predicted by PlantCare.

Other things that need to be added have been added in this article.

Reviewer 2 Report

This revised version answered most of my concerns. There is still one main concern I feel the authors need a little more attention. Although the authors claimed that "Univariate comparisons were analyzed for significance. There is no comparison between multiple factors, so significance is not shown in the figure", the word "significantly" was used multiple times in the manuscript. Only when the statistical analysis was performed can the word "significantly" be used to describe the difference. I still suggest authors consider this issue carefully.

Author Response

This article has been modified where significance is concerned